# Response Mechanisms of “Hass” Avocado to Sequential 1–methylcyclopropene Applications at Different Maturity Stages during Cold Storage

**DOI:** 10.3390/plants11131781

**Published:** 2022-07-05

**Authors:** Daniela Olivares, Miguel García-Rojas, Pablo A. Ulloa, Aníbal Riveros, Romina Pedreschi, Reinaldo Campos-Vargas, Claudio Meneses, Bruno G. Defilippi

**Affiliations:** 1Instituto de Investigaciones Agropecuarias, INIA-La Platina, Santa Rosa 11610, Santiago 8831314, Chile; olivaresdaniela@gmail.com (D.O.); miguel.garcia@inia.cl (M.G.-R.); pablo.ulloa@inia.cl (P.A.U.); 2ANID—Millennium Science Initiative Program—Millennium Nucleus for the Development of Super Adaptable Plants (MN-SAP), Santiago 8370186, Chile; anibal.riveros.o@gmail.com (A.R.); claudio.meneses@uc.cl (C.M.); 3Escuela de Agronomía, Facultad de Ciencias Agronómicas y de los alimentos, Pontificia Universidad Católica de Valparaíso, Calle San Francisco s/n, La Palma, Quillota 2260000, Chile; romina.pedreschi@pucv.cl; 4Centro de Estudios Postcosecha, Facultad de Ciencias Agronómicas, Universidad de Chile, Santa Rosa 11315, Santiago 8820808, Chile; reinaldocampos@uchile.cl; 5Departamento de Fruticultura y Enología, Facultad de Agronomía e Ingeniería Forestal, Pontificia Universidad Católica de Chile, Av. Vicuña Mackenna 4860, Santiago 7820436, Chile; 6Departamento de Genética Molecular y Microbiología, Facultad de Ciencias Biológicas, Pontificia Universidad Católica de Chile, Av. Libertador Bernardo O’Higgins 340, Santiago 8331150, Chile

**Keywords:** avocado, 1–methylcyclopropene, ethylene, RNA-seq, quality, cold storage

## Abstract

1–Methylcyclopropene (1–MCP) is used for extending the postharvest life of the avocado during storage. Evaluated the effect of 1–MCP application at different times after harvest, i.e., 0, 7, 14, and 21 d at 5 °C, to identify the threshold of the ethylene inhibition response in “Hass” avocado. Our results showed that fruits from two maturity stages at harvest: low dry matter (20–23%) and high dry matter (27%). Changes in ethylene production rates and transcript accumulation of genes involved in ethylene metabolism were measured at harvest and during storage. 1–MCP treated fruit up to 14 d of storage showed similar values of firmness and skin color as fruit treated at harvest time. In contrast, when the application was performed after 21 d, the fruit showed ripening attributes similar to those of the untreated ones. To further understand the molecular mechanisms responsible for the lack of response to 1–MCP at 21 d of storage, transcriptomic analysis was performed. Gene ontology analyses based on the DEG analysis showed enrichment of transcripts involved in the ‘response to ethylene’ for both maturity stages. All genes evaluated showed similar expression profiles induced by cold storage time, with a peak at 21 d of storage and an increased softening of the fruit and peel color. This was a two-year field study, and results were consistent across the two experimental years. Our results should help growers and markets in selecting the optimal timing of 1–MCP application in “Hass” avocados and should contribute to a deeper understanding of the molecular mechanisms of the avocado ripening process.

## 1. Introduction

Chile is one of the main producers and exporters of “Hass” avocado for long–distant markets such as North America, Europe, and Asia in the world [1,2]. A major challenge for the Chilean industry is to provide homogenous fruits in terms of quality and ripening attributes, even after several days of shipping and storage (20–60 d) [1,3,4,5,6]. Avocado fruits have several physiological characteristics that limit their storage life, such as high respiration and ethylene production rates at the onset of ripening [7,8]. In global trade, avocado fruit holding in terms of “time in the system” is an integral aspect of shipping to distant markets [9]. Low–temperature storage (4−7 °C) in combination with a controlled atmosphere (CA) are the main technologies used to extend the storage life [10,11]. Storage in CA resulted in better ripe fruit quality than storage in air and delaying the establishment of CA on “Hass” avocado did not affect the skin color development during storage, the time taken to ripen after storage, or the incidence of rots in the ripe fruit [12]. Temperature is an important variable during fruit growth, and maturation may also influence fruit quality by either hastening or delaying horticultural maturity [13]. Ethylene inhibitors, such as 1-methylcyclopropene (1–MCP), are broadly used in climacteric fruits to delay ripening and irreversibly bind to ethylene receptors on cell membranes and have shown several benefits in extending postharvest life [14,15,16]. In avocado, the beneficial effects of 1–MCP have been shown in different varieties and can be summarized as (i) a reduction in the softening rate, (ii) a diminish in the color change rate, and (iii) a reduction decrease in susceptibility to physiological disorders [17,18,19]. However, despite these beneficial effects during storage, the application of 1–MCP in “Hass” avocado could cause a detrimental delay in the ripening progress and abnormal overall ripening at specific maturity stages. These effects are characterized by inhibition of pulp softening, uneven ripening, and delayed color development [14,20,21]. Similar effects were observed in Packham’s pear [3,22,23]. These effects result in extensive shelf–life periods (>15–20 d at 20 °C) to reach a ready-to-eat stage, causing excessive water loss from the fruit. The response to 1–MCP at low-temperature storage is affected by several factors, such as maturity stage, delay in application after harvest and fruit source.

Avocado fruit, like other climacteric fruit, shows induction of a variety of genes during ripening [24]. These genes are associated with cellulase, ethylene metabolism, and cytochrome P–450 oxidase [25]. 1–aminocyclopropane–1–carboxylate synthase 1 (*PamACS1)* mRNA increased and peaked prior to the climacteric peak, whereas overexpression of 1–aminocyclopropane–1–carboxylate oxidase (*PamACO*) mRNA levels occur at the climacteric peak of ethylene production. A base level of one ethylene response sensor (*PamERS1*) transcript was detected at harvest. However, the *PamERS1* transcript was induced at the climacteric peak of ethylene production [24]. *PamACS1,* 1–aminocyclopropane–1–carboxylate synthase 2 (*PamACS2*), and *PamACO* transcript levels in avocado increased markedly during ripening, with a detectable expression of avocado *PamACS* and *PamACO* genes and very low activity of ACS and ACO at harvest that increased in correlation with the beginning of the climacteric rise during ripening [24]. Ethylene receptor 1 (*PamETR*) and *PamERS1* transcripts were upregulated at the onset of normal ripening and correlated with the level of climacteric ethylene production.

The inhibitor 1–MCP reduces the effects of ethylene and/or interferes with its function in plants, blocking the transduction signal cascade that leads to the expression of genes related to the ethylene response [26]. 1–MCP in avocado inhibited the transcription of *PamACS1* and suppressed *PamACO* and *PamERS1* mRNA to trace levels [24]. Low temperatures during storage stimulate *PamACO* and *PamACS* gene expression in apples and pears [27,28,29], and low temperatures in the orchard cause dramatic induction of avocado fruit ripening with parallel increases in ethylene biosynthesis and expression of receptor genes [30].

The present study aimed: (i) to investigate the effect of sequential 1–methylcyclopropene (1–MCP) applications during cold storage on physiological parameters related to ripening and on changes in transcript accumulation of specific genes involved in ethylene metabolism (synthesis, perception, and signaling) and (ii) to elucidate the molecular mechanisms of 1–MCP on postharvest fruit ripening of avocado through a transcriptomics analysis of the differences between responsive and non-responsive fruits.

## 2. Results

### 2.1. Ripening Related Physiological Parameters

#### 2.1.1. Ripening Parameters

Flesh firmness and skin color are the main quality parameters showing the ripening progress of “Hass” avocado, and they also determine the acceptability of fruit batches by consumers [1,4,20]. For flesh firmness, as expected after storage, there was a significant delay in pulp softening in 1–MCP-treated fruit in comparison with untreated control fruit (UTC; Figure 1A). The UTC in high dry matter (HDM) showed a firmness value close to 49 N, which corresponds to a decrease of 82% of the flesh firmness observed at harvest (263 N), which was very close to the value measured at the ripe stage (ready-to-eat stage). The 1–MCP-treated avocados, independent of application time, stayed firm (>200 N) during cold storage and started to soften only after removal to shelf storage temperature (20 °C). For 1–MCP-treated avocados, fruit treated either at harvest or after 7 and 14 d of cold storage showed similar values to those observed at harvest in both maturity stages (272 N in low dry matter (LDM) and 263 N in HDM). On the other hand, only when the 1–MCP application was performed after 21 d of cold storage did the flesh firmness at the end of storage decrease by 26.1% compared to that at harvest. The same results were observed for both maturity stages (LDM and HDM).

After 35 d of cold storage, avocado fruit were exposed to 20 °C until the flesh firmness reached levels indicative of the ripe stage (10–20 N; ready-to-eat stage). As previously observed, the untreated control fruit softened rapidly, reaching the ready-to-eat stage in 3 d for LDM and 4 d for HDM. LDM treated with 1–MCP fruit required 10–14 d to reach the ready-to-eat stage after cold storage (35 d at 5 °C). HDM treated with 1–MCP fruit instead required 7–10 d to reach the same conditions (Figure 1B).

Skin color did not show major changes during cold storage. The 1–MCP treated fruits (up to day 14) showed a delay in color development. However, when the inhibitor was applied after 21 d of storage, no significant differences were observed compared with the UTC fruit (Figure 1C). At the ripe stage at the end of shelf life at 20 °C, both the fruit treated at 21 d and the fruit without application reached an advanced level according to the color scale (data not shown).

#### 2.1.2. Physiological Parameters

The ethylene production rate of untreated control avocados (UTC) was higher for LDM fruit after 35 d of cold storage (Figure 2A). LDM untreated fruit showed an ethylene production rate 15 times higher than 1–MCP treated fruit, independent of the application time. HDM untreated control fruit showed a lower level of ethylene production than LDM fruit. These lower levels could be due to the advanced ripening stage observed after 35 d of storage. UTC fruit (without 1–MCP application) showed a higher respiration rate than 1–MCP treated fruit, which was almost double that of the LDM fruit (Figure 2B). The 1–MCP treatments also resulted in an increase over respiration rate for both dry matter contents dependent on the 1–MCP application time, which was significant mainly for the LDM fruit. The results obtained for the ripening and physiological parameters showed the expected delay in ripening due to the application of the ethylene inhibitor (1–MCP), reaching the ripe stage later compared to the untreated avocado for both evaluated maturity stages. Interestingly, the closer to harvest when the application of 1–MCP was applied, the greater the delay in ripening; and if the application was delayed more than 21 d from harvest, avocado fruit did not respond to the ethylene perception inhibitor.

### 2.2. Fruit Characteristics at the 1–MCP Application Time

To understand the response of the fruit to the 1–MCP ethylene perception inhibitor, ripening and physiological parameters were also examined and transcriptomic analyses were performed on the avocados immediately before each 1–MCP application (control fruit).

#### 2.2.1. Ripening Parameters

After harvest, for both dry matter (DM) contents, the fruit showed a slight reduction in flesh firmness for up to 14 d of storage at 5 °C (Figure 3A). However, after 21 d storage, the fruit showed a lower firmness, which was more evident in LDM fruit with a 56.6% decrease compared to that at harvest time. For HDM fruit, this reduction reached 26% with respect to firmness at harvest. Similarly, skin color stayed green (level 1) during the first 14 d of storage and showed an increase after 21 d for both DM contents (Figure 3B).

#### 2.2.2. Physiological Parameters

The ethylene production rate in “Hass” avocado stored for 21 d at 5 °C was significantly higher than in fruit at harvest and after storage for 7 and 14 d (Figure 3C). Additionally, higher ethylene production was observed in fruits with LDM. Similar to the change in ethylene levels, a lower respiration rate was observed with up to 14 d of cold storage, increasing after 21 d of storage and reaching higher values than the rate of fruit at harvest (Figure 3D).

#### 2.2.3. Transcriptomic Analysis

To further understand the lower response to 1–MCP at 21 d, an RNA sequencing experiment was carried out. The samples correspond to fruit before each application. Based on the above results, this analysis was performed considering three ripening stages: (i) the optimum 1–MCP application time (harvest), (ii) the latest stage responsive to ethylene inhibition (14 d storage), and (iii) fruit that did not respond to inhibition during storage (21 d storage).

A total of 1,126,168,980 high-quality paired-end reads were obtained with a Q25 of 99% for all libraries (Appendix A). Approximately 93.1% of the total trimmed reads mapped to the reference genome of *Persea americana* var. *drymifolia* [31]. A group of 1358 differentially expressed genes (DEGs) between 0 and 14 d at 5 °C in the LDM fruits were identified (FDR < 0.001 and logFC > |2| < logFC), corresponding to 701 upregulated and 657 downregulated genes (Table 1 and Appendix A). For the comparison of 0 vs. 21 d at 5 °C, a greater number of DEGs were identified (4480 DE), corresponding to 1829 upregulated and 2651 downregulated DEGs (Table 1 and Appendix A). In the HDM samples, we observed a pattern similar to that of LDM fruits. For the comparison of 0 vs. 14 d, 1121 DEGs were identified, corresponding to 477 upregulated and 644 downregulated genes (Table 1 and Appendix A). A group of 5279 DEGs between 0 and 21 d at 5 °C was identified, corresponding to 2214 upregulated and 3065 downregulated genes (Table 1 and Appendix A).

To test whether the DEGs were significantly linked to specific biological processes, gene set enrichment analysis was conducted with gene ontology (GO) annotation terms using the ClusterProfiler package of R. Significant GO terms revealed numerous functions associated with ripening, such as cell wall biogenesis, which could affect flesh firmness. Other relevant functions were related to the phenylpropanoid metabolic process and secondary metabolite biosynthetic process, which favors the development of color in the epidermis of the avocado and other characteristic quality parameters indicative of the ripening process. Both functions associated with the cell wall and skin color are related to the biological process category ´response to ethylene´. The ‘response to ethylene’ category showed a slight increase in enrichment at 21 d of cold storage in LDM fruit (Figure 4; Appendix A). This increase was more significant in the HDM fruit, where this was the only biological process category enriched after 21 d of cold storage (Figure 5; Appendix A).

#### 2.2.4. Expression Patterns of Genes Involved in Ethylene Biosynthesis

In this work, the genes that encode *PamACS1*, *PamACS2*, *PamACO*, *PamETR1*, *PamCTR1*, *PamEIL1*, and *PamEIL3* and their expression patterns were analyzed in “Hass” avocado by semiquantitative qPCR immediately before each 1–MCP application. In the case of the genes involved in ethylene synthesis, *PamACS1* and *PamACO* presented a similar expression profile, which was influenced by the cold storage period, with the highest peak observed at 21 d, and non-significant differences were observed between LDM and HDM fruit (Figure 6A,C). For *PamACS2*, the LDM avocado presented an expression pattern similar to those previously described, with a peak at 21 d of storage. Contrary to that observed in the HDM fruit, a decrease in expression was observed at 21 d compared to day 0 (Figure 6B). In the case of the ethylene signaling genes, *PamETR1*, *PamCTR1*, and *PamEIL3* presented similar expression profiles induced by cold storage time, with a peak at 21 d of storage and non-significant differences were observed between LDM and HDM fruits (Figure 7A,B,D). *PamEIL1* expression was similar during cold storage, and there were no differences between LDM and HDM fruit (Figure 7C).

## 3. Discussion

Ripening and physiological parameters were examined to characterize ethylene inhibition response to sequential applications of 1–methylcyclopropene in “Hass” avocado during cold storage. The ethylene action inhibitor 1–MCP has been found to be effective in overcoming the effects of ethylene in a range of perishable fruits [32]. However, this effect depends on the application time, i.e., fruit maturity stage, of the inhibitor. 1–MCP was reported to delay ripening, as expressed in changes in flesh firmness, skin color, ethylene production, and respiration rate in “West Indian” cultivars and in “Hass” avocado fruit [17,33]. After 35 d storage at 5 °C, UTC (without 1–MCP) fruits softened rapidly, reaching a ready-to-eat stage in 3–4 d, unlike 1–MCP-treated fruit, which required 7–14 d to be ready to eat, being softer than fruit treated with 1–MCP independent of application time or maturity stage (harvest time) and taking more days to reach the ready-to-eat stage, and this effect was earlier described by Woolf et al. [22]. Flesh firmness retention was significantly enhanced in response to 1–MCP treatment, consistent with the fact that softening is one of the most obvious ripening processes in response to ethylene. Similar effects of 1–MCP in delaying avocado fruit softening have been previously observed for “Ettinger,” “Fuerte,” and “Hass” [21,22,33,34,35,36]. However, a decrease in flesh firmness was observed in the fruit treated with 1–MCP at 14 and 21 d compared to that at harvest time, which would indicate that the fruit did not respond to 1–MCP applied after 21 d after harvest. At different 1–MCP application times (14 and 21 d) for both maturity stages, the fruit showed a reduction in flesh firmness, which was more evident after 21 d. Flesh firmness is one of the quality parameters that reflects the progress of ripening (maturity), decreasing with the increase in the cold storage period (in regular air). This response was observed at 14 and 21 d after cold storage, mainly caused by an advance in the ripening of the fruit [37]. In addition, flesh firmness, and changes in other quality attributes, such as skin color, were also observed in fruit under cold storage after treatment with 1–MCP. This change was evident after 21 d of cold storage, and the fruit had still failed to reach maturity. The UTC fruit presented a greater development in color than the treated fruit independent of harvest time, but there was no delay for the fruit treated at 21 d with 1–MCP compared to the UTC.

In the present study, 1–MCP treatment significantly delayed the onset of climacteric ethylene production and respiratory patterns in all treatments were indicative of a more delayed ripening in LDM fruit compared to HDM. These results are complementary to those of other studies on avocado [17,38,39] and other climacteric fruit, such as banana [40,41], apple [42], and apricot [43].

Low temperatures during storage stimulate *PamACO* and *PamACS* gene expression in apples and pears [27,28,29], and low temperatures in the orchard cause dramatic induction of avocado fruit ripening with parallel increases in ethylene biosynthesis and expression of receptor genes [7].

To understand the metabolic pathways and biological processes that occur in fruit after a period of 14 and 21 d of cold storage, differentially expressed genes were used to perform GO analyses using comparisons of 14 d versus harvest and after 21 d versus harvest of LDM fruit; and 14 d versus harvest and after 21 d versus harvest of HDM fruit. A greater number of up–and downregulated genes were observed in the comparison of 21 d versus harvest of HDM fruits, mainly associated with the advance of fruit maturity, which generates activation of a greater number of enzymes and genes related to metabolic processes, including ‘cell wall biogenesis’ (GO: 0042546), ‘monocarboxylic acid metabolic process’ (GO: 0032787), ‘intracellular signal transduction’ (GO: 0035556), ‘polysaccharide metabolic process’ (GO:0005976), ‘response to ethylene’ (GO:0009723), ‘phenylpropanoid metabolic process’ (GO:0009698), and ‘organic acid catabolic process’ (GO: 0016054), among others.

Within the context of this research, the ripening of climacteric fruit, such as avocado, involves a series of coordinated metabolic events that affect the fruit morphology, biochemistry, physiology, and gene expression [44], and these alterations modify many quality attributes, such as the skin color, texture, and flavor of the fruit [45], which are mainly regulated by ethylene.

The ethylene synthesis pathway is well established in higher plants [46], and regulatory control is achieved in two steps: the formation of 1–aminocyclopropane –1–carboxylic acid (ACC) from S-adenosyl–L–Met and the conversion of this intermediate to ethylene [47]. The first step is catalyzed by the enzyme ACC synthase (ACS), and the second is catalyzed by ACC oxidase (ACO). In addition, the ethylene receptor is an upstream element that plays a negative regulatory role in the ethylene signal transduction pathway. The receptor proteins encoded by each gene have different structures and levels of expression, and five ETR1-like genes, ETR1 [48], ERS1 [49], ETR2 [50], EIN4, and ERS2 [50], have been identified in *Arabidopsis* plants. All enzymes are encoded by multigene families that generate multiple points of control at which ethylene synthesis may be regulated [47]. In the present study, cold storage induced the expression levels of all genes, which were much higher after 21 d at 5 °C, concomitant with the higher levels of ethylene and an increased softening of the fruit, without showing a difference between maturity stages. On the other hand, the application of 1–MCP induced delayed softening and ethylene and CO_2_ production during cold storage and downregulated the expression of genes involved in ethylene biosynthesis and ethylene action. Similar effects of 1–MCP in delaying ripening have been observed in various avocado cultivars [10,14,17,22] and other species, such as banana [41], pear [51], and pineapple [52].

## 4. Materials and Methods

### 4.1. Materials

The study was carried out in a commercial orchard located at Cabildo (Chile; 32°24′46″ S 70°54′26″ W) during two consecutive growing seasons (2017/2018 and 2018/2019) with similar meteorological conditions (Appendix A). “Hass” avocado fruit were harvested at two maturity stages based on their dry matter content: low dry matter (LDM) close to 20–23% and high dry matter (HDM) exceeding 27%. For each maturity stage, four hundred avocado fruit were harvested. The harvest condition was based on sampling 20 fruit for dry matter analysis as commercially practiced. Fruit were immediately transported to the laboratory facilities at the Instituto de Investigaciones Agropecuarias (INIA, Santiago, Chile).

### 4.2. Dry Matter (DM)

DM content of each avocado fruit was determined by drying the mesocarp samples (skinless biopsies) in an oven at 103 °C until a constant weight [53]. For the analysis, two mesocarp biopsies (5 mm diameter taken at the Equatorial zone) were performed for each fruit and then sealed with petroleum jelly and wax. The avocado fruit were identified and characterized by their DM content according to Pedreschi et al. [54].

### 4.3. 1–MCP Applications

Avocado fruits were randomly selected and arranged for the different applications of 1–MCP1 (SmartFresh^TM^, Agrofresh, Santiago, Chile). Applications were made at different times during storage: harvest and after a period of cold exposure: 7 d, 14 d, and 21 d, respectively. 1–MCP was applied in a sealed container (600 L) at 300 ppb for 12 h at 5 °C. For each treatment, 40 avocado fruit were packaged in ventilated plastic boxes and stored at 5 °C and 80% relative humidity for 35 d. Fruit without 1–MCP application were used as untreated control (UTC) under the same storage conditions. Additionally, fruit were evaluated at each application time (0 d, 7 d, 14 d, and 21 d) after cold storage and when fruit reached a ready-to-eat stage (mesocarp firmness < 8.9 N).

### 4.4. Mesocarp Firmness

Mesocarp firmness was measured with a penetrometer (Effegi, FT327, FT011, Alfonsine, Italy) equipped with a 4 (for firm fruit) or 8 (for ready-to-eat stage) mm plunger tip. The measurements were made on each avocado fruit (*n* = 5 for treatment) side at 20 °C. Firmness measurements were performed at harvest, after cold storage, and at the end of shelf life. The skin was removed before firmness evaluation, and mesocarp firmness was expressed as Newton (N) [55].

### 4.5. Skin Color

Skin color was assessed visually using a hedonic scale (from 1 to 5) according to Rivera et al. [4], where 1 = 100% of the peel surface is green; 2 = 20% of the peel surface is colored black/purple (violet) on green; 3 = 60% of the peel surface is colored black/purple (violet) on green; 4 = 100% of the peel surface is purple (violet), and 5 = 100% of the peel surface is black.

### 4.6. Respiration and Ethylene Production Rates

The respiration and ethylene production rates were individually measured on ten avocado fruit per treatment before 1–MCP application and after cold storage (35 d at 5 °C). The respiration rate at 20 °C was measured by injecting 1 mL of gas from the headspace into a gas analyzer (PBI-Dansensor Checkmate 9900, Ringsted, Denmark), and values were expressed as mL CO_2_ kg^−1^ h^−1^ [56]. For measurement of ethylene, fruit were placed in a 1.6 L plastic container and sealed for 2 h at 20 °C, thus avoiding the excessive accumulation of carbon dioxide [57]. Then, 1 mL of gas was taken from the headspace and injected into a gas chromatograph (Shimadzu GC 8A, Tokyo, Japan) as described by Olivares et al. [57]. Values were expressed as μL C_2_H_4_ kg^−1^ h^−1^.

### 4.7. Bioinformatic Analysis

#### 4.7.1. RNA-Seq Experiment

Total RNA from avocado mesocarp was isolated using a modified hot borate method [58]. The quantity and quality of the RNA were assessed with a Qubit^®^ 2.0 fluorometer (InvitrogenTM by Life Technologies, Singapore) by measuring the A260/280 ratio and by electrophoresis on a 1.2% formaldehyde agarose gel. Total RNA was assessed by a fragment Analyzer Automated CE System (Advanced Analytical Technologies, Inc., Ames, IA, USA), and 1 ug from each sample was used as input for the Illumina TruSeq RNA HT sample preparation commercial kit, according to the manufacturer’s instructions. Eighteen samples without application 1–MCP [(two maturity stages (LDM and HDM) × three application times (0, 14, and >21 d) × three biological replicates)] were analyzed by sequencing using the HiSeq 4000 platform (Illumina) with 2 × 100 bp (paired-end mode) reads.

#### 4.7.2. Sequencing Data Analysis

The sequencing quality of each library was analyzed by FastQC v0.11.8 software (Babraham Bioinformatic, Cambridge, England). After quality analysis, the trimming process was carried out by Trim galore v0.6.2 software (Babraham Bioinformatic, Cambridge, England) to remove sequence adapters, and reads were filtered by sequence quality with the default parameters. Following this procedure, trimmed reads were mapped to the reference genome of *Persea americana* var *drymifolia* [31] through STAR v2.7.3a. software (Cold Spring Harbor Laboratory, New York, USA) [59].

#### 4.7.3. Differential Gene Expression Analysis

Differential expression analysis was performed by DESeq2 v 1.26.0 (Bioconductor, Parkville, Australia) using filter parameters of an FDR < 0.001 and fold change (FC) > 2, comparing differentially expressed genes between both fruit without 1–MCP in different application times (14 d and >21 d at 5 °C) from the LDM and HDM stages and the control fruit (harvest). Further analysis, including gene ontology (GO) enrichment analysis, was performed with the ClusterProfiler package in R with functions enrichGO and simplified using an FDR > 0.05 and redundancy semantic of 0.7 (cut-off value).

### 4.8. cDNA Synthesis and Real-Time Quantitative PCR Assays

First-strand cDNA was obtained by reverse transcription with 2 ug of total RNA (isolated for RNA-seq experiment, Section 4.7.1) as the template using M-MLV reverse transcriptase (Promega, Madison, WI, USA) and oligo dT primers according to standard procedures.

Seven ethylene metabolism genes were selected to characterize ethylene synthesis: [*PamACO* (1–aminocyclopropane–1–carboxylate oxidase, GenBank M32692.1); *PamACS1* (1–aminocyclopropane–1–carboxylate synthase 1, GenBank AF500119.1); *PamACS2* (1–aminocyclopropane–1–carboxylate synthase 2, GenBank AF500120.1)], perception and signaling [*PamETR1* (Ethylene Receptor 1), *PamCTR1* (Constitutive Triple Response 1), *PamEIL1* (Ethylene Insensitive 3-like 1) and *PamEIL3* (Ethylene Insensitive 3-like 3)]. The selection of these genes was carried out based on a previous study [11,30,60]. The abundance of each transcript was analyzed by real–time PCR with a LightCycler Real–Time PCR System (Roche Diagnostics, Mannheim, Germany) using SYBR Green™ as a fluorescent dye to measure the amplified DNA products derived from the RNA. qPCR was performed with four replicates, and the gene expression values were normalized to the *PamTCPB* gene (T–complex protein 1 subunit beta, GenBank KT246107). Relative expression levels were determined using the comparative Ct method of Pfaffl [61], which was designed to measure the accumulation of mRNA relative to that of a constitutively expressed reference gene. Primers (Appendix A) were designed using the Primer Premier 5.0 software package (Premier Biosoft International, Palo Alto, CA, USA) and were synthesized by IDT (Integrated DNA Technologies, Coralville, IA, USA).

### 4.9. Experimental Design and Statistical Analysis

The experiments were performed with a completely randomized design. Prior to analyzing percentages, data were arcsine transformation, with non-transformed values used for presentation in figures. The data were subjected to statistical analyses of variance (ANOVA), and the means were separated by a least significant difference (LSD) test at 5% significance using the statistical software InfoStat (Version 2015, Universidad Nacional de Córdoba, Córdoba, Argentina).

## 5. Conclusions

The results from our study demonstrated that the delayed application of 1–MCP during storage is effective if applied before 21 days of cold storage; after this period, the fruit does not respond to the treatment. As already shown in the literature, the application of 1–MCP during cold storage causes a delay in the ripening process of avocado due to an inhibition of ethylene metabolism, mainly by decreasing expression levels of genes related to critical enzymes of ethylene synthesis, *PamACS1*, *PamACS2* and *PamACO* and of the receptors *PamETR1* and *PamCTR1*. However, this study showed that this delay, at least at the transcriptional level, lasted close to 21 d after harvest, and at this stage, transcripts involved in ethylene biosynthesis/perception increased their expression, concomitant with a decrease in maturity and physiological attributes. Results from this study could be very useful for the avocado industry in selecting the optimal timing of 1–MCP application in “Hass” avocado and also in providing a deeper understanding of the mechanisms of ethylene metabolism for novel technologies based on the sequential release of 1–MCP during storage from pads or films already available.

## Figures and Tables

**Figure 1 plants-11-01781-f001:**
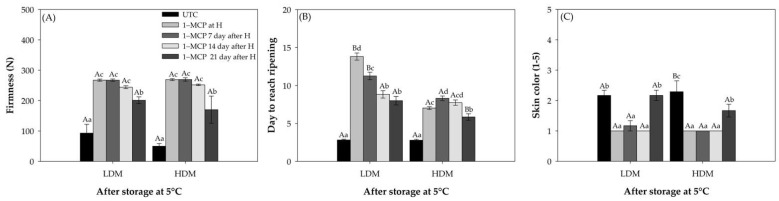
Quality and physiological parameters of “Hass” avocado. Untreated control (UTC, fruit without 1–MCP), application of 1–MCP at harvest, 7 d, 14 d, and >21 d after harvest, all fruit were put in cold storage (35 d at 5 °C) in regular air. (**A**) Firmness (N). (**B**) Days to reach ripening. (**C**) Skin color (range 1–5). Different lowercase and capital letters indicate a significant difference among treatments and maturity stages by LSD analysis (*p* < 0.05). H: Harvest.

**Figure 2 plants-11-01781-f002:**
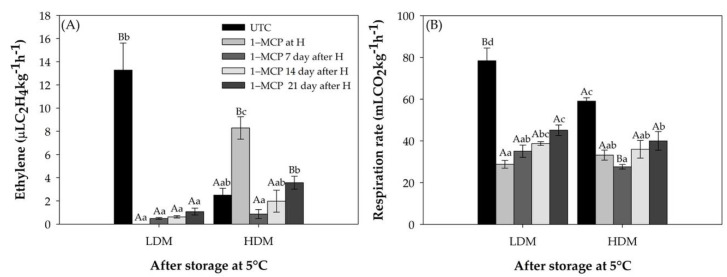
Ethylene production and respiration rate of “Hass” avocado. Untreated control (UTC, fruit without 1–MCP), application of 1–MCP at harvest, 7 d, 14 d, and >21 d after harvest, all fruit were put in cold storage (35 d at 5 °C) in regular air. (**A**) Ethylene production (μL C_2_H_4_ kg^−1^ h^−1^). (**B**) Respiration rate (mL CO_2_ kg^−1^ h^−1^). Different lowercase and capital letters indicate a significant difference among treatments and maturity stages by LSD analysis (*p* < 0.05). H: Harvest.

**Figure 3 plants-11-01781-f003:**
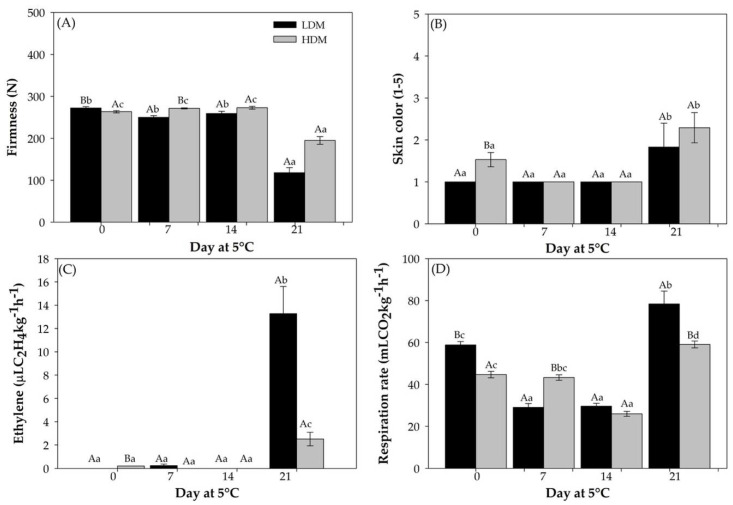
Quality and physiological parameters in “Hass” avocado in the moment of applying 1–MCP. Fruit from LDM and HDM were stored for 0, 7, 14, and >21 d at 5 °C in regular air. (**A**) Firmness (N). (**B**) Skin color (range 1–5). (**C**) Ethylene production (μL C_2_H_4_ kg^−1^ h^−1^). (**D**) Respiration rate (mL CO_2_ kg^−1^ h^−1^). Different lower-case and capital letters indicate a significant difference among treatments and harvest time by LSD analysis (*p* < 0.05).

**Figure 4 plants-11-01781-f004:**
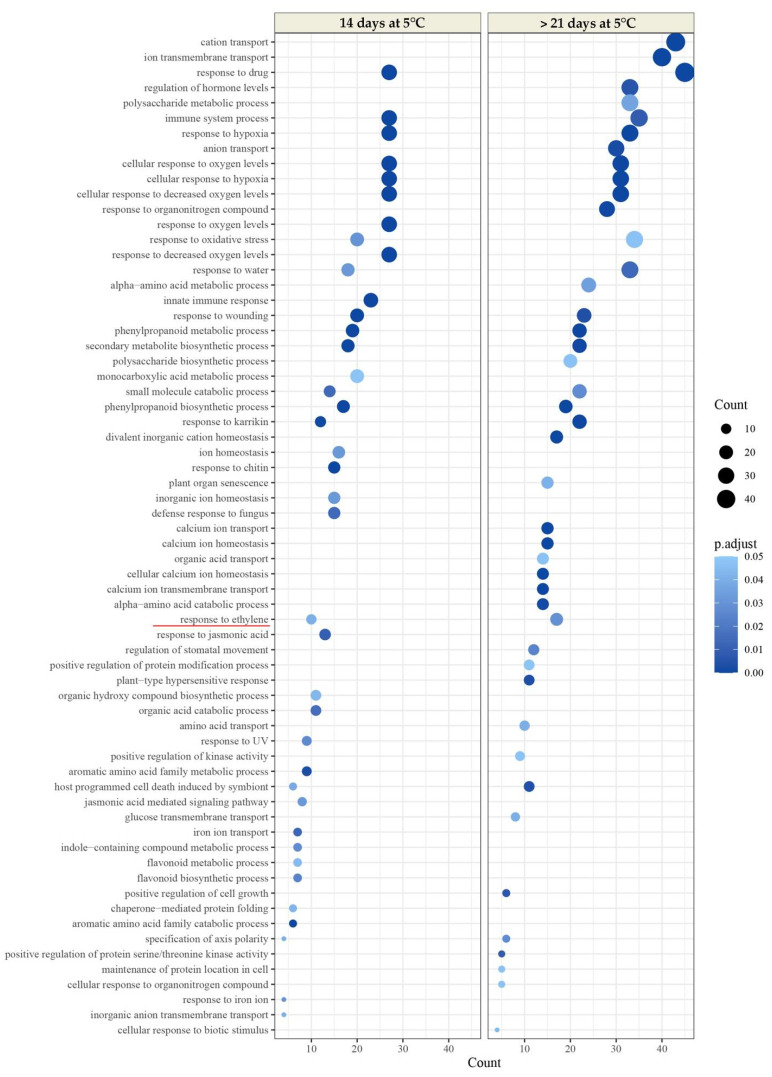
Dot plot of enriched Gene Ontology (GO) terms for the genes upregulated by cold storage compared with fruit at harvest of LDM. The *Y*-axis indicates the GO term, and the *X*-axis shows the count of genes per GO term. The input upregulated genes were 587 and 892 for 14 d at 5 °C and >21 d at 5 °C, respectively. The color gradient indicates the adjusted *p*-value (0.05) using the Benjamin-Hochberg method. The GO term ‘response to ethylene’ (GO: 0009723) is marked in a red line (left).

**Figure 5 plants-11-01781-f005:**
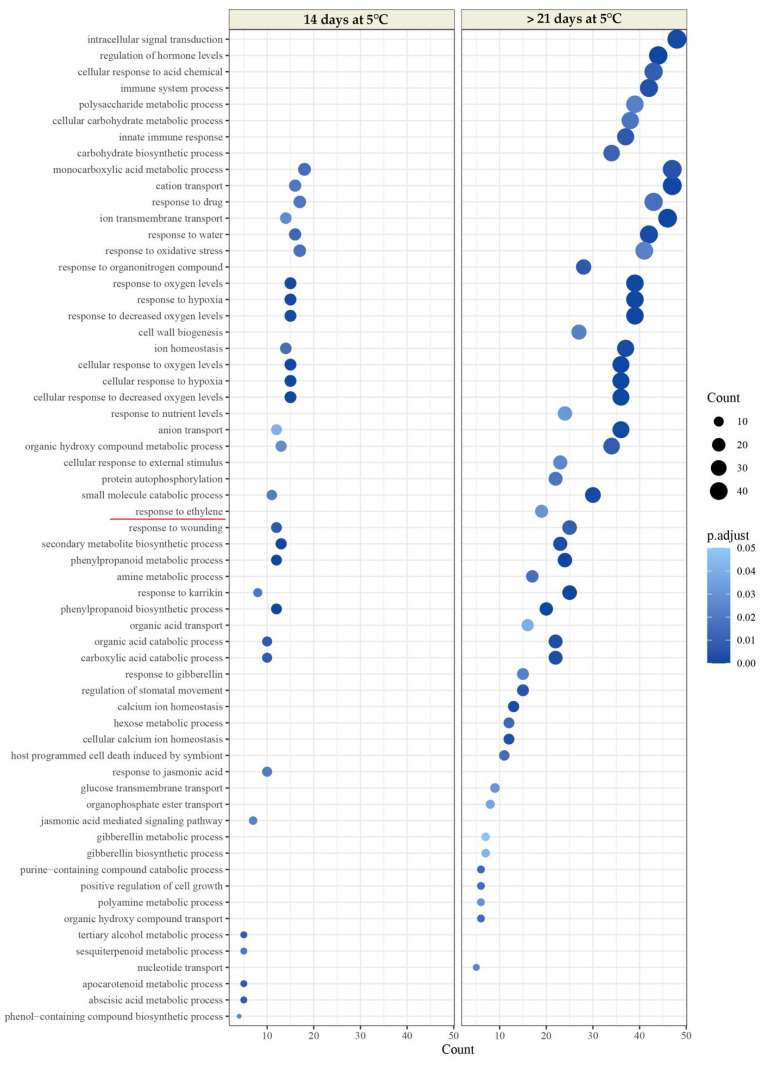
Dot plot of enriched Gene Ontology (GO) terms for the upregulated genes by cold storage compared with fruit at harvest of HDM. The *Y*-axis indicates the GO term, and the *X*-axis shows the count of genes per GO term. The input upregulated genes were 356 and 1428 for 14 d at 5 °C and >21 d at 5 °C, respectively. The color gradient indicates the adjusted *p*-value (0.05) using the Benjamin-Hochberg method. The GO term ‘response to ethylene’ (GO: 0009723) is marked in a red line (left).

**Figure 6 plants-11-01781-f006:**
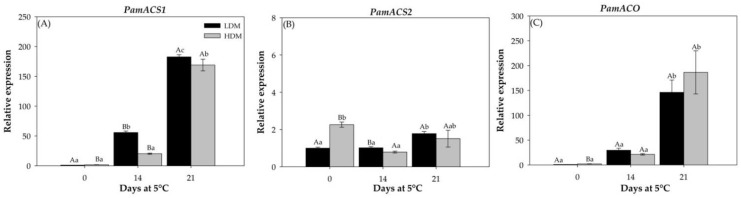
Relative expression analyses of ethylene synthesis genes in “Hass” avocado at the moment of 1–MCP application. (**A**) Relative expression of *PamACS1*, (**B**) Relative expression of *PamACS2*. (**C**) Relative expression of *PamACO*. The transcript accumulation of genes was determined by qPCR. Expression was normalized to the housekeeping gene *PamTCPB*, and the expression is relative to that in LDM fruit (day 0). Bars represent the means of three replicates ± SE. Different lowercase and capital letters indicate a significant difference among treatments and maturity stages by LSD analysis (*p*-value < 0.05).

**Figure 7 plants-11-01781-f007:**
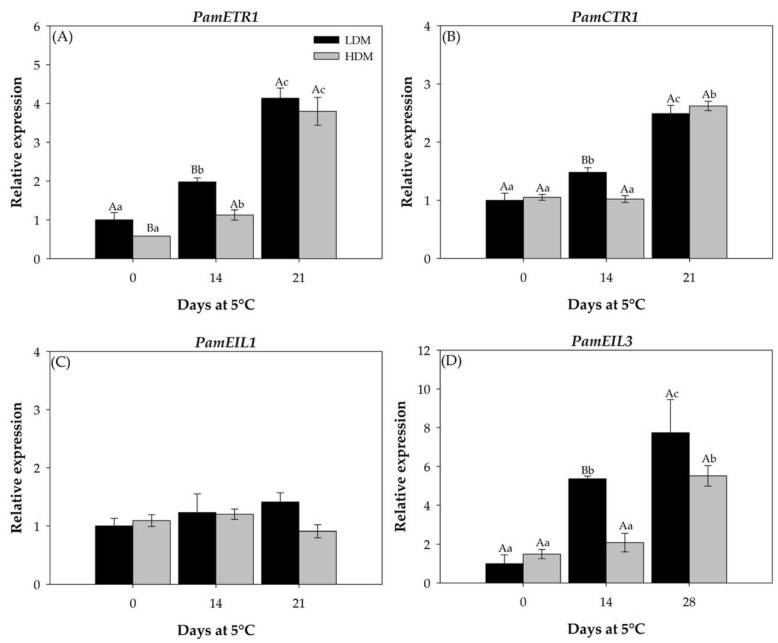
Relative expression analyses of genes from ethylene signaling pathway in “Hass” avocado before application of 1–MCP. (**A**) Relative expression of *PamETR1*, (**B**) Relative expression of *PamCTR1*. (**C**) Relative expression of *PamEIL1*. (**D**) Relative expression of *PamEIL3*. The transcript accumulation of genes was determined by qPCR. Expression was normalized to the housekeeping gene *PamTCPB* and the expression is relative to that in LDM fruit (day 0). Bars represent the means of three replicates ± SE. Different lowercase and capital letters indicate a significant difference among treatments and harvest times by LSD analysis (*p*-value < 0.05).

**Table 1 plants-11-01781-t001:** The number of differentially expressed genes (DEGs) with different cold storage times compared with fruit at harvest.

Harvest	Trial	Up	Down
LDM	14 d at 5 °C vs. Harvest	701	657
	>21 d at 5 °C vs. Harvest	1829	2651
HDM	14 d at 5 °C vs. Harvest	477	644
	>21 d at 5 °C vs. Harvest	2214	3065

LDM: Fruit harvested at 20–23% dry matter; HDM: Fruit harvested at 27% dry matter; FDR < 0.001 and logFC > |2| < logFC; d: day.

## Data Availability

Data sharing is not applicable.

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
