# Peer review of "Response Mechanisms of “Hass” Avocado to Sequential 1–methylcyclopropene Applications at Different Maturity Stages during Cold Storage"

_plants, 2022, doi:10.3390/plants11131781_

Round 1
Reviewer 1 Report
Olivares et al. deal with the effect of application time of 1-MCP on avocado fruit quality attributes during postharvest storage. Experiments were well-conducted, and the paper is very well written. I advise the authors to deal with the following comments within the manuscript rather than the response letter.
1. Line 3 (title): Is the abbreviation really needed here?
2. Lines 22/ 254: the storage time/ physiological (maturity) stage/ ethylene perception/ ethylene production/ ethylene sensitivity threshold? Which of those is relevant?
3. Lines 20-32: please mention this is a two-year field study, and results were consistent across the two experimental years
4. Abstract & discussion: Why fruit a different maturity stage at harvest respond the same way to 1-MCP application? It may be expected that both ethylene production and sensitivity are different between the two stages. Nevertheless, it may also be that both stages under study are well below the climacteric phase.
5. Line 43: controlled atmosphere was not evaluated in the current study. It would be interesting to discuss the implications of the obtained findings in fruit exposed to modified atmosphere, as done in commercial practice
6. Line 44: Temperature is far the most important factor during the postharvest chain period (Tsaniklidis et al. 2021 Postharvest Biol Technol 179, 111586). Non-optimal temperature regimes adversely affect quality attributes and postharvest longevity.
7. Lines 53-54: Were these disadvantages noted here?
8. Line 62: Please see a relevant study including DEG analysis in another climacteric fruit (Zhang et al., 2021 Postharvest Biol Technol 180, 111622)
9. Line 83: aimed to
10. Line 258: why the delay in application did not have any effect?
11. Lines 263-264: is this delay positive or negative? Please see lines 56-58
12. Lines 273-274: This firmness loss typically comes as a result of modifications in cell wall structure (Fanourakis et al. 2022 Agronomy 12, 778). For instance, pectin solubilization occurs, which is mediated via cell wall hydrolytic enzymes (e.g., galactosidases, and cellulases). Attenuating the activity of enzymes related to softening during the postharvest period can very well improve shelf life by postponing firmness loss.
13. Abstract: it needs to be mentioned that despite the promotive effects of 1-MCP (timely) application, a delay in ripening is apparent. (Please see lines 56-58)
14. Line 295: please avoid using symbols (> and vs.) within the sentence. Please use full words.
15. Line 333: The plural of fruit is fruit when referring in the same species
16. Line 328: interesting that you used fruit from two different seasons. Are there any meteorological data (light intensity, temperature, relative air humidity)? If yes, please show the means of the two seasons in supplementary material
17. Line 356: why did you use different probe diameter? Can you still compare the results, when a different probe diameter is employed?
18. Line 356: in the text it is commonly mentioned as flesh firmness. However, what was assessed is fruit firmness, since the skin was not removed before evaluation. Please adjust.
19. Lines 21, 186: please avoid the royal “we”
20. The weakest point of the manuscript is the discussion. Authors may exercise more attention in highlighting what is new as compared to previous findings, and what are the implications for commercial practice? Suggestions for future work (e.g. include sensory panels, modified atmosphere etc.)
Reviewer 2 Report
The study is quite interesting. The results have both practical applications and scientific merit. Refer to the Word file for suggested edits.

Reviewer 3 Report
The manuscript entitled: "Response mechanisms of “Hass” avocado under 1 methylcyclopropene (1-MCP) application at different maturity stages during cold storage", is adequately written, the research design is appropriate, the methods are adequately described and the results clearly presented, however it offers no new information and no new slant on the topic, it is lacking in originality. There are many papers published containing the same information of the present work (Amin et al., 2014; Cabia et al., 2014; Hershkovitz et al., 2005; Meyer and Terry, 2010; Olivares et al., 2020; Woolf et al., 2005). Thus I recommend the rejection of the manuscript.
Amin, IAH, Poerwanto, R, Kartika, JG, 2014. 1-MCP application to prolong avocado shelflife. Acta Horticulturae. 1120: 131-136.
Cabia, NC, Daiuto, ER, Vieites, RL, Smith, RE, 2014. Maintaining the Quality and Antioxidant Capacity of 'Hass' Avocados after Applying 1-Methylcyclopropene (1-MCP) 4 (3): 233-240.
Hershkovitz, V, Saguy, SI, Pesis, E, 2005. Postharvest application of 1-MCP to improve the quality of various avocado cultivars. POSTHARVEST BIOLOGY AND TECHNOLOGY 37 (3): 252-264.
Meyer, MD, Terry, LA, 2010. Manipulating the Ripening of Imported Avocado 'Hass' Fruit during Cold Storage Using e plus (R) Ethylene Remover or 1-Methylcyclopropene (1-MCP). Acta Horticulturae 858: 295-300.
Olivares, D, Alvarez, E, Véliz, D, García-Rojas, M, Díaz, C, Defilippi, B., 2020. Effects of 1-Methylcyclopropene and Controlled Atmosphere on Ethylene Synthesis and Quality Attributes of Avocado cvs. Edranol and Fuerte. Journal of Food Quality. 2020. 1-14.
Woolf, AB, Requejo-Tapia, C, Cox, KA, Jackman, RC, Gunson, A, Arpaia, ML, White, A, 2005. 1-MCP reduces physiological storage disorders of 'Hass' avocados. POSTHARVEST BIOLOGY AND TECHNOLOGY 35(1): 43-60.
Round 2
Reviewer 1 Report
authors deal with my comments upon their convenience
no motivation to provide new ones
Reviewer 3 Report
The authors have highlighted the originality of this study respect the previous works.This is a study that deserves to be published.